# Association of polycyclic aromatic hydrocarbons in moss with blood biomarker among nearby residents in Portland, Oregon

Igor Burstyn[1]*, Geoffrey H. Donovan[2], Yvonne L. Michael[1], Sarah Jovan[2]

**1** Dornsife School of Public Health, Drexel University, Nesbitt Hall, Philadelphia, Pennsylvania, United States of America, **2** USDA Forest Service, PNW Research Station, Portland, Oregon, United States of America

* igor.burstyn@drexel.edu

## Abstract

Polycyclic aromatic hydrocarbons (PAHs) are air pollutants that are costly to measure using traditional air-quality monitoring methods. We used an epiphytic bio-indicator (moss genus: *Orthotrichum*) to cost-effectively evaluate atmospheric deposition of PAHs in Portland, Oregon in May 2013. However, it is unclear if measurements derived from these bioindicators are good proxies for human exposure. To address this question, we simultaneously, measured PAH-DNA adducts in blood samples of non-smokers residing close to the sites of moss measurements. We accounted for individual determinants of PAH uptake that are not related to environmental air quality through questionnaires, e.g., wood fires, consumption of barbecued and fried meats. Spearman rank correlation and linear regression (to control for confounders from the lifestyle factors) evaluated the associations. We did not observe evidence of an association between PAH levels in moss and PAH-DNA adducts in blood of nearby residents (e.g., all correlations p≥0.5), but higher level of adducts were evident in those who used wood fire in their houses in the last 48 hours. It remains to be determined whether bio-indicators in moss can be used for human health risk assessment.

## 1. Introduction

Polycyclic aromatic hydrocarbons (PAHs) are a class of air pollutants, some of which are linked to adverse health outcomes [1, 2]. PAHs are by-products of the incomplete combustion of organic matter, including tobacco products, fossil fuels and firewood, and are abundant in smoked, fried, or grilled food. Risk management requires understanding pathways of exposure, best done though comparing environmental measurements that reflect different sources to measurements of internal dose through biomarkers. Cost-effective methods to accurately evaluate exposures to PAH are lacking, unlike that for total particle matter, typically employed in air pollution research and regulations.

Lichen and moss are among the most commonly used bio-indicators of atmospheric PAHs and can accumulate both particle and gas-phase PAHs [3, 4]. Their leaves and lobes lack cuticles, allowing PAHs to diffuse easily into cells [5] and a high-surface area traps particles [6]. In

**Data Availability Statement:** Data contain potentially identifying or sensitive information, and therefore under ethical clearance that permitted conduct of the study and as part of informed

consent, we are not able to share that data, IRB Protocol Number 1402002647. However, there are some circumstances under which we can be compelled to share data. Such requests should be made to the Human Research Protection Program (HRPP) that is responsible for providing administrative and regulatory support to the Drexel University Institutional Review Boards (IRB), via email hrpp@drexel.edu.

**Funding:** The author(s) received no specific funding for this work.

**Competing interests:** I, Igor Burstyn, have read the journal's policy and the authors of this manuscript have the following competing interests: Igor Burstyn is an expert witness in a litigation that concerns matters related to use of lichen and moss as indicators of human exposure and risk. This does not alter our adherence to PLOS ONE policies on sharing data and materials. All other authors have declared that for them no competing interests exist.

addition, lichen and moss do not have roots, making them dependent on atmospheric sources of nutrients and water and, therefore, bioaccumulation reflects airborne sources of PAHs.

Lichens and moss contain levels of mid-to-high molecular weight PAHs that correlate with nearby stationary air-quality monitors in studies conducted by different groups across diverse locations [3, 6, 7]. Jovan et al. [8] demonstrated that moss can be used to cost-effectively map the distribution of PAHs across residential areas of Portland, Oregon, which correlated with both traffic and tree canopy cover; earlier related research on metals indicated the same moss species, *Orthotrichum lyellii* [9], can be used to identify sources of such pollution [10]. It is uncertain how PAH levels in lichens and moss relate to doses received by humans. To increase our understanding of this critical question, we conducted a follow-up study to Jovan et al. (2021) investigating the relationship between levels of PAHs in moss and that of PAH-DNA adducts in the blood of nearby residents.

## 2. Materials and methods

### 2.1 Study area and sampling strategy

Portland is a city in northwest Oregon, USA; *Orthotrichum lyellii* is the only species of moss or lichen that is common across a wide variety of sites in Portland. Participants were non-smokers residing with non-smokers recruited though an email list distributed to employees of US Forest Service's Pacific Northwest Research Station located in Portland. Smokers self-identified as is appropriate in human health research among trusted volunteers who had no reason to conceal if they smoked. We did not ask about health because there is no *a priori* reason why health would impact levels of PAH in blood and moss on properties, thereby introducing confounding. We collected moss samples in May of 2013 on hardwood trees located on properties of study participants, from a height of at least 1 meter to reduce the influence of spray from cars and from dog urine. We recorded the location of each sample point using a high-accuracy GPS (Garmin Oregon 450). Drexel University Institutional Review Board approved the study protocol (IRB ID 1402002647), which included written informed consent.

### 2.2 Human biological monitoring and questionnaires

To capture recent exposure to wood fires that would affect levels of biomarkers, participants completed questionnaires that capture demographics and exposure to known sources of PAH during the preceding 48 hours: wood fires in house, consumption of fried, barbecued, grilled, or smoked food. We recognize that 48 hours is shorter than half-life of PAH but it is a time period that is typically used in epidemiology to more reliably elicit person's typical behaviors, with a recall on the order of months being unreliable; however, it is customarily assumed that behaviors in the last 48 hours are segregate persons who typical exhibit these behaviors over long periods of time versus those who do not.

A licensed phlebotomist collected 50 ml of blood by venipuncture. Blood samples were transported by overnighted delivery for quantification of PAH-DNA adduct at the laboratory at Columbia University's Mailman School of Public Health following protocol of Santella et al. [11–13].

### 2.3 Moss sample preparation and testing

Methods of moss sampling and testing are reported in detail by Jovan et al. [8] and are only briefly summarized here. Immediately after collection, we stored samples at 4 ˚C in metalized polyester Kapak bags. On a sterilized lab bench, we removed debris and trimmed off most brown tissue. On average, samples were stored for one week or less before analysis.

We ground samples in the lab, using an ASE 200 (Dionex, Sunnyvale, CA) for pressurized liquid extraction of PAHs. We loaded moss samples into stainless steel cells, with 1 gram of Florisil$^{®}$ (activated magnesium silicate; U.S. Silica Company, Berkeley Springs, USA) placed at the outlet of the cell. We used dichloromethane for PAH extraction. We pre-heated cells to 100 ˚C for five minutes then processed with two static extraction cycles. Between cycles, we flushed the cells with 50% of cell volume solvent. At the end of cycles, the cell was flushed with high purity nitrogen gas (N2). We concentrated the PAH extract under N2 at ambient temperature to a final volume of 1mL.

We analyzed the extract using a method similar to EPA 8270D, utilizing an Agilent 6890GC with 5973 mass selective detector, operating in full scan mode, and a Restek Rxi-XLB column. For quality assurance, we ran a method blanks, and spiked each sample with isotopically labeled (deuterated) PAH surrogates that served as internal standards to monitor calibration and performance of the GC-MS instrument. In quality control experiments, average percent recoveries of known concentrations of isotopically labeled surrogate PAH compounds added to moss samples were all indistinguishable from 100%, within one standard deviation; all method blanks were at or below detection (the details of QA/QC are already published as Appendix A of Jovan et al., 2021).

## 2.4 Statistical analysis

We replaced all non-detectable samples values with half of the limit of detection [14]. We estimated Spearman rank correlation of measurements of PAH in moss and blood adducts that were detected. We explored the correlation structure among PAH levels in moss via principal components analysis (PCA) of detected levels of PAH and used it to limit the number of PAH in moss that need to be considered in relation to PAH-DNA adducts. Briefly, PCA identifies clusters of PAHs that are correlated among themselves which can be due to common sources, and such clusters are independent of each other and explain the maximum total variability in the data [15]. The number of principal components that we considered in analysis was indicated by examination of the scree plot. PAHs representative of each selected principal component i.e., those with the highest loadings, were used as predictors of blood PAH-DNA adduct levels. Because PAH-DNA were right-skewed (appeared to be log-normal), we applied logarithmic transformation to ensure validity of parametric statistical tests that require assumption of Gaussian distribution. Associations of PAH-DNA adducts in the blood were related to PAH levels in moss using linear regression controlling for wood burning in the house and consumption of smoked, grilled, or barbecued foods, as well as race (white vs. other), age (continuous in years) and gender; all covariates were selected *a priori*.

## 3. Results

We recruited 53 non-smoking volunteers for the study, for whom we collected blood samples and moss samples from their property. The volunteers were predominantly female (n = 31), white (n = 47) and aged on average 51 years (range 30 to 68). Only 3 reported wood fires on their properties and 17 reported eating fried or grilled food within 48 hours of the measurements.

The level of PAH-DNA adducts and PAH levels in moss are summarized in Table 1. Among the 32 measurements of adducts that were above the detection limit (5.6 adducts/$10^8$nucleotides), the average level was 11.8 adducts/$10^8$nucleotide with a range of 7.9 to 22.9; coefficient of variation (CV) of 3.2/11.8 = 27%. PCA found that two principal components accounted for 71% of the correlation structure among PAHs in moss, each associated with PAHs typically occurring in the particle and gas phase in the air, respectively. Based on

**Table 1. Descriptive statistics of measurements the levels of polycyclic aromatic hydrocarbons (PAH) in moss and blood among measurements above limit of detection; 53 samples collected for each measurement.**

| Measurement | Number detected | Mean | Standard Deviation | Median | Minimum | Maximum |
|---|---|---|---|---|---|---|
| *PAH blood adduct (/$10^8$ nucleotides)* | 32 | 11.8 | 3.2 | 11.5 | 7.9 | 22.9 |
| *PAH in moss (µg/kg)* | | | | | | |
| Acenaphthene | 28 | 4.1 | 4.0 | 3.2 | 1.77 | 23.5 |
| Acenaphthylene | 25 | 4.2 | 4.2 | 3.0 | 1.7 | 23 |
| Anthracene | 49 | 3.9 | 1.5 | 3.7 | 1.67 | 10.3 |
| Benzo[*a*]anthracene | 53 | 20.0 | 17.0 | 16.3 | 6.29 | 124 |
| Benzo[*a*]pyrene | 42 | 15.0 | 6.7 | 15.1 | 3.82 | 31.3 |
| Benzo[*b*]fluoranthene | 53 | 22.6 | 10.7 | 21.3 | 3.33 | 66.5 |
| Benzo[*g,h,i*]perylene | 49 | 22.7 | 9.8 | 21.1 | 7.625 | 68.5 |
| Benzo[*k*]fluoranthene | 52 | 15.9 | 5.3 | 15.3 | 6.57 | 29.6 |
| Chrysene | 53 | 38.6 | 12.8 | 36.1 | 14.2 | 65.6 |
| Dibenzo[*a,h*]anthracene | 14 | 10.4 | 4.8 | 9.1 | 4.525 | 20.6 |
| Dibenzofuran | 48 | 6.0 | 7.0 | 4.5 | 2.37 | 41.8 |
| Fluoranthene | 53 | 34.4 | 11.1 | 32.7 | 13.4 | 66.7 |
| Fluorene | 19 | 4.8 | 5.3 | 2.9 | 1.8 | 20.5 |
| Ideno[1,2,3-*cd*]pyrene | 35 | 11.0 | 4.7 | 9.9 | 3.23 | 25.7 |
| 1-Methylnaphthalene | 47 | 6.0 | 8.3 | 3.9 | 1.48 | 46.9 |
| 2-Methylnaphthalene | 53 | 6.6 | 9.5 | 5.3 | 2.32 | 72.4 |
| Naphthalene | 53 | 13.1 | 16.3 | 9.8 | 3.89 | 97.7 |
| Perylene | 49 | 16.6 | 6.3 | 16.2 | 2.43 | 39.5 |
| Phenanthrene | 53 | 20.9 | 5.7 | 21.4 | 10.1 | 35.6 |
| Pyrene | 53 | 28.5 | 10.1 | 29.1 | 10.75 | 68.4 |

correlations/loading with the principal components, we selected benzo[*a*]pyrene, pyrene, and naphthalene to further investigate associations with PAH-DNA blood adducts in linear regressions. The details of correlation coefficients and PCA are presented in the Supporting Information S1 File, which contains all the supporting tables and figures from the raw SAS outputs of the supporting calculations.

The absolute levels of PAH in moss do not have any particular meaning due to absence of reference ranges, but it is noteworthy that two different PAHs representative of gas- and particle-phases were detectable in all samples (namely: naphthalene and pyrene, respectively). The more volatile gas-phase naphthalene was more variable (CV of 16.3/13.1 = 124%) than the less volatile solid-phase pyrene (CV 10.1/28.5 = 35%). Benzo[*a*]pyrene, an established carcinogen, was detected in 42/53 = 79% of measurements. There was little evidence of either rank or linear correlation between any of the individual PAHs in moss and PAH-DNA adducts (details not shown).

Linear regression models with logarithm of PAH-DNA adduct as the dependent variable did not reveal any associations with PAH in moss. The unadjusted associations are illustrated in the Fig 1. Adjustment for age, race, gender, wood fires or dietary history did not substantially alter conclusion drawn from the Fig 1. The regression model containing all covariates and all three PAHs in moss had $R^2$ = 0.2 with no indication of violation of the assumption of the analyses in the residual plots (not shown). The only notable association was consistent excess of PAH-DNA adducts in relation to wood fires in the house in preceding 48 hours, by a factor of about two (p = 0.1). However, this was based on only three observations: in presence of wood fires, average level of PAH-DNA adduct was 12.9 *vs*. 8.0 adducts/$10^8$ nucleotided in absence of reported fires.

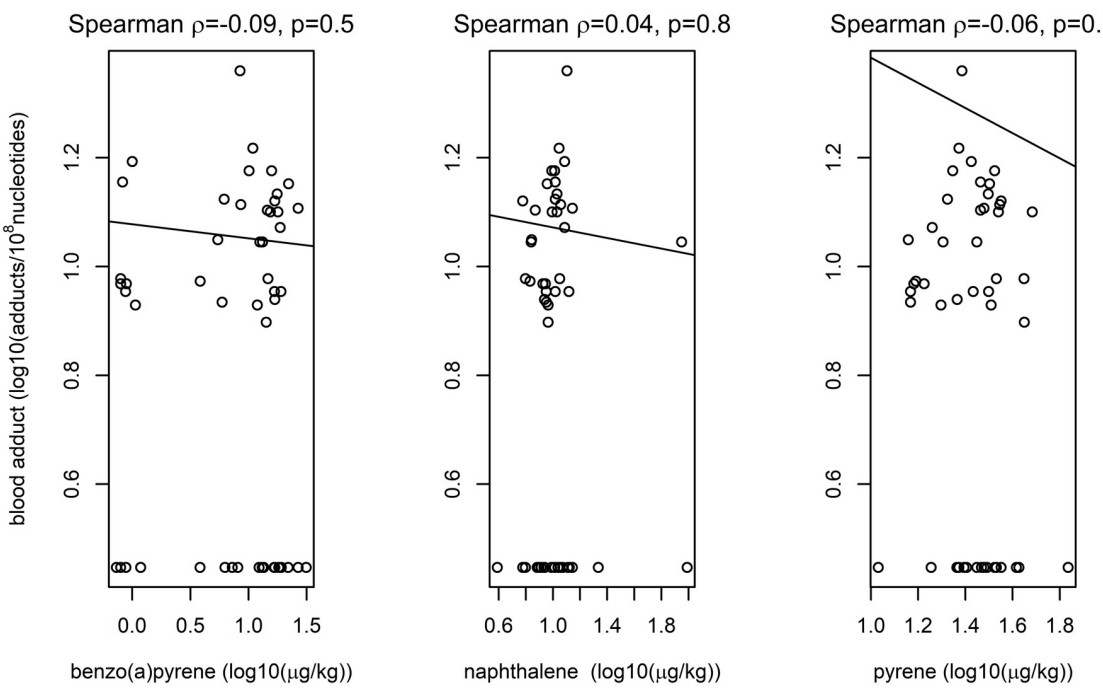

**Fig 1. Linear models fit to logarithms of PAHs in moss and their PAH-DNA blood adducts along with rank correlations; non-detects of adducts and benzo[*a*] pyrene were imputed as half of the limit of detection.**

## 4. Discussion

We did not observe evidence to support the hypothesis that levels of PAH in the moss *O. lyellii* were related to levels of these exposures among nearby residents. Among the strengths of our approach was selection of non-smoking volunteers who resided with non-smokers, thereby controlling for a major source of confounding by design. That PAH-DNA adducts were potentially sensitive to at least one known source of PAH (wood fires) suggests these biomarkers may be useful for identifying more highly exposed individuals [13, 16]. However, overall, we did not find that PAH levels in moss helpful in human exposure assessment in the studied setting.

Part of the reason for poor correlation of PAHs in moss and their adducts in human blood may be that the moss PAH concentrations were greatly influenced by sampling conditions (e.g., daily humidity and type of tree, local traffic conditions), whereas PAH-DNA adduct in human blood are subject to "physiological dampening" [17] that is less sensitive to these external conditions that can change daily (even if they are relevant). This is supported by our finding that the CV of adducts was lower than those of PAH measurements in moss. We attempted to account for some of these sources of confounding by limiting sampling to only one month and but we were not able to control for factors that Jovan et al. [8] saw as important to variation of PAH content of moss in the area (i.e., daily humidity and type of tree, local traffic conditions). The levels of PAH in moss collected in May 2013 that we observed in the current study are substantially lower than those obtained using identical methods in Portland in December 2013 by Jovan et al. [8], underscoring the importance of seasonal trends and spatial variability. Perhaps better results can be obtained with long-term (multiple measures) monitoring of moss on properties.

Our study had several limitations. This study required relatively large moss samples (median cleaned and dried sample weight: 9.1 grams). While this was not a major challenge in

Portland's temperate climate, collecting samples of this size in other areas could be challenging. More sensitive laboratory techniques may overcome this limitation. The different half-lives of PAHs in moss and human PAH-DNA adducts were one of the major limitations of our study. While PAH-DNA adducts have half-lives of about 3–4 months [18], the half-lives of PAHs in moss are not known. One study of temporal variability related particle-bound PAHs (such as benzo[*a*]pyrene and pyrene) in lichens to levels in the air two months prior [3] although due to their lower lipid content, Knulst et al. [19] argue that moss is expected to represent shorter timeframes. If the two reservoirs of PAH (DNA adducts and moss content) integrate these levels over different timeframes, then a correlation would not be expected. This remains an important area of research that may help calibrate moss measurements to biomarkers in humans. The degree of use of wood fires can vary for the same binary response, introducing random covariate misclassification error into our results that would tend to bias estimates of any impact of wood fires in regression models.

## 5. Conclusion

Our research does not support the use of moss bio-indicators to assess risk of human exposure to PAH in Portland, Oregon. Although we do not challenge the established observation that PAH in environment can cause adverse health effects and produce adducts, our results indicate that it is difficult to demonstrate this relationship with bioindicators in the studied setting. It is plausible that in other environments with a stronger relationship between personal exposure and contamination of properties by PAH bioindicators will prove advantageous over established methods of assessing risk and exposure to humans.

## Supporting information

**S1 File. All the supporting tables and figures from the raw SAS outputs of the principal components analysis.**
(PDF)

## Acknowledgments

Thanks to Dr. Todd Rosentiel and Dr. Kenneth Stedman, Portland State University, for generously providing lab space. Consultation on PAH methods and extraction services provided by Specialty Analytical.

## Author Contributions

**Conceptualization:** Igor Burstyn, Geoffrey H. Donovan, Yvonne L. Michael, Sarah Jovan.

**Data curation:** Geoffrey H. Donovan, Sarah Jovan.

**Formal analysis:** Igor Burstyn.

**Methodology:** Igor Burstyn, Geoffrey H. Donovan, Yvonne L. Michael, Sarah Jovan.

**Project administration:** Geoffrey H. Donovan.

**Visualization:** Igor Burstyn.

**Writing – original draft:** Igor Burstyn.

**Writing – review & editing:** Igor Burstyn, Geoffrey H. Donovan, Yvonne L. Michael, Sarah Jovan.

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
