## [Decision Letter · Decision Letter 0]

10 Aug 2022

PONE-D-22-13009Association of polycyclic aromatic hydrocarbons in moss with blood biomarker among nearby residents in Portland, OregonPLOS ONE

Dear Dr. Burstyn,

Thank you for submitting your manuscript to PLOS ONE. After careful consideration, we feel that it has merit but does not fully meet PLOS ONE’s publication criteria as it currently stands. Therefore, we invite you to submit a revised version of the manuscript that addresses the points raised during the review process.

We look forward to receiving your revised manuscript.

Kind regards,

Fung-Chi Ko, PhD

Academic Editor

PLOS ONE

Journal Requirements:

a) Did participants provide their written or verbal informed consent to participate in this study?

6. We note you have included a table to which you do not refer in the text of your manuscript. Please ensure that you refer to Table 1 in your text; if accepted, production will need this reference to link the reader to the Table

Additional Editor Comments:

Reviewer’s comments:

Although inconclusive, the manuscript entitled "Association of polycyclic aromatic hydrocarbons in moss with blood biomarker among nearby residents in Portland, Oregon," is interesting and may be useful to other researchers. Yet some gaps need to be addressed before publication. One issue which needs to be addressed stems from surveys of volunteers asking about possible exposure within 48 h hours. This short period is not explained or discussed, although PAHs-adduct half-lives are estimated to be 3-4 months. The authors state that half-lives of PAHs in moss are unknown. However, one would expect, based on PAH stability, they will be at least on a month scale that is much longer than 48h hours.

It appears the authors omitted the description of sample preparation and analysis of DNA adducts and which specific adducts were analyzed. Alternatively, the reference to the lab performing the work should be provided if this was done as a service. Furthermore, some of the material reported in the results belong to the experimental section. The results section is surprisingly short, extended by the lengthy, simply formatted table. Nevertheless, the authors further narrowed it by focusing on only three representatives PAHs.

The major revision of the manuscript needed to address these issues and detailed comments below.

Detailed comments

Line 19. Description of PAHs – which is a full class of compounds as "component" is awkward; please rephrase perhaps as follows "Polycyclic aromatic hydrocarbons (PAHs) are air pollutants.".

The direct speech in the abstract using "we" pronoun is refreshing but somewhat overused. To make articles easier to read, some sentences should be rewritten. This is also the case for other sections of the paper.

Line 30. It should be "adducts" in the plural.

Line 76. In this section, the authors carefully describe IRB protocol and data collection. The questionnaire included information of possible exposure to PAHs for 48h. Yet the question is if this is sufficient time for correlations made. The PAHs are quite stable lipophilic compounds that can bioaccumulate. Thus investigation to relationships with longer times seems to be more appropriate. Perhaps authors need to add some explanation for this or can include in the discussion how their approach is appropriate.

Lines 84, 90. The symbols for degree Celsia are inconsistent; please ensure correct symbols are used everywhere. Usually, "space" is in front of the degree symbol.

Line 93. Use N2 for nitrogen gas.

Line 96. "Mass selective detector" does not need to be capitalized. Also, provide mass range and ionization – presumably electron ionization.

Line 101. Consider switching the order of 1st two sentences – The current presentation of the 1st sentence without specification of type of statistical analysis is confusing.

The experimental section does not address analysis of the PAH-adduct markers. Some material from Lines 118-126 including figure 1 belongs to the experimental section.

Lines 128-131 also belong to the experimental section, supporting the study's quality but not being actual results.

Line 137-139 describes how the study was narrowed, yet correlation for other PAHs could be performed and included in the expanded figure or supplementary materials. Such data would add strength to the paper.

Line 140. The authors mention that specific values for concentration are not informative due to the lack of data on PAHs in moss. It seems it would be useful to compare these results to papers referred in the introduction, justifying the use of this approach.

Table 1 need significant revision to achieve a professional look, but also concerning content. PAHs naming nomenclature needs to be revised (there are numerous excellent resources on correct naming). The alphabetic order of PAHs is confusing; it would be better to organize PAH based on volatility or the number of rings and also provide their totals. Correlations with totals of these groupings could be insightful and give a different look at attempted correlation.

Reviewers' comments:

Reviewer's Responses to Questions

**Comments to the Author**

1. Is the manuscript technically sound, and do the data support the conclusions?

Reviewer #1: No

Reviewer #2: Partly

2. Has the statistical analysis been performed appropriately and rigorously? 

Reviewer #1: No

Reviewer #2: Yes

3. Have the authors made all data underlying the findings in their manuscript fully available?

Reviewer #1: No

Reviewer #2: No

4. Is the manuscript presented in an intelligible fashion and written in standard English?

Reviewer #1: No

Reviewer #2: Yes

5. Review Comments to the Author

Reviewer #1: The possible relationship between atmospheric PAHs and ones in human blood. This is not clear yet, therefore this study did not conclude any relationship between bioindicator and human blood. This result is common and expected. If the authors show evidences for this unclear circumstance, it would be orijinal for the readers.

Reviewer #2: Although inconclusive, the manuscript entitled "Association of polycyclic aromatic hydrocarbons in moss with blood biomarker among nearby residents in Portland, Oregon," is interesting and may be useful to other researchers. Yet some gaps need to be addressed before publication. One issue which needs to be addressed stems from surveys of volunteers asking about possible exposure within 48 h hours. This short period is not explained or discussed, although PAHs-adduct half-lives are estimated to be 3-4 months. The authors state that half-lives of PAHs in moss are unknown. However, one would expect, based on PAH stability, they will be at least on a month scale that is much longer than 48h hours.

It appears the authors omitted the description of sample preparation and analysis of DNA adducts and which specific adducts were analyzed. Alternatively, the reference to the lab performing the work should be provided if this was done as a service. Furthermore, some of the material reported in the results belong to the experimental section. The results section is surprisingly short, extended by the lengthy, simply formatted table. Nevertheless, the authors further narrowed it by focusing on only three representatives PAHs.

The major revision of the manuscript needed to address these issues and detailed comments below.

Detailed comments

Line 19. Description of PAHs – which is a full class of compounds as "component" is awkward; please rephrase perhaps as follows "Polycyclic aromatic hydrocarbons (PAHs) are air pollutants.".

The direct speech in the abstract using "we" pronoun is refreshing but somewhat overused. To make articles easier to read, some sentences should be rewritten. This is also the case for other sections of the paper.

Line 30. It should be "adducts" in the plural.

Line 76. In this section, the authors carefully describe IRB protocol and data collection. The questionnaire included information of possible exposure to PAHs for 48h. Yet the question is if this is sufficient time for correlations made. The PAHs are quite stable lipophilic compounds that can bioaccumulate. Thus investigation to relationships with longer times seems to be more appropriate. Perhaps authors need to add some explanation for this or can include in the discussion how their approach is appropriate.

Lines 84, 90. The symbols for degree Celsia are inconsistent; please ensure correct symbols are used everywhere. Usually, "space" is in front of the degree symbol.

Line 93. Use N2 for nitrogen gas.

Line 96. "Mass selective detector" does not need to be capitalized. Also, provide mass range and ionization – presumably electron ionization.

Line 101. Consider switching the order of 1st two sentences – The current presentation of the 1st sentence without specification of type of statistical analysis is confusing.

The experimental section does not address analysis of the PAH-adduct markers. Some material from Lines 118-126 including figure 1 belongs to the experimental section.

Lines 128-131 also belong to the experimental section, supporting the study's quality but not being actual results.

Line 137-139 describes how the study was narrowed, yet correlation for other PAHs could be performed and included in the expanded figure or supplementary materials. Such data would add strength to the paper.

Line 140. The authors mention that specific values for concentration are not informative due to the lack of data on PAHs in moss. It seems it would be useful to compare these results to papers referred in the introduction, justifying the use of this approach.

Table 1 need significant revision to achieve a professional look, but also concerning content. PAHs naming nomenclature needs to be revised (there are numerous excellent resources on correct naming). The alphabetic order of PAHs is confusing; it would be better to organize PAH based on volatility or the number of rings and also provide their totals. Correlations with totals of these groupings could be insightful and give a different look at attempted correlation.

6. PLOS authors have the option to publish the peer review history of their article (what does this mean?). If published, this will include your full peer review and any attached files.

Reviewer #1: No

Reviewer #2: No

---

## [Author Response · Author response to Decision Letter 0]

12 Sep 2022

Comments of the Editor (ED)

ED2. Please amend your current ethics statement to address the following concerns:

a) Did participants provide their written or verbal informed consent to participate in this study?

RESPONSE: We obtained written consent; study protocol was approved by Drexel University IRB (this was already in the original submission on line 72). The expanded sentence reads, with the new text highlighted: “Drexel University Institutional Review Board approved the study protocol (Protocol number 1402002647), which included written informed consent.”

ED3. Please include your full ethics statement in the ‘Methods’ section of your manuscript file. In your statement, please include the full name of the IRB or ethics committee who approved or waived your study, as well as whether or not you obtained informed written or verbal consent. If consent was waived for your study, please include this information in your statement as well.

RESPONSE: We made the requested change; see response to previous comment.

ED4. In your Data Availability statement, you have not specified where the minimal data set underlying the results described in your manuscript can be found. PLOS defines a study's minimal data set as the underlying data used to reach the conclusions drawn in the manuscript and any additional data required to replicate the reported study findings in their entirety. All PLOS journals require that the minimal data set be made fully available. For more information about our data policy, please see http://journals.plos.org/plosone/s/data-availability.

RESPONSE: We are prohibited from sharing data by IRB clearance that governed the project. We do not have permission to share any parts of the data. Copy of IRB approval is attached a material not for external reviewers.

ED5. We note that Figure 1 in your submission contain [map/satellite] images which may be copyrighted. All PLOS content is published under the Creative Commons Attribution License (CC BY 4.0), which means that the manuscript, images, and Supporting Information files will be freely available online, and any third party is permitted to access, download, copy, distribute, and use these materials in any way, even commercially, with proper attribution. For these reasons, we cannot publish previously copyrighted maps or satellite images created using proprietary data, such as Google software (Google Maps, Street View, and Earth). For more information, see our copyright guidelines: http://journals.plos.org/plosone/s/licenses-and-copyright.

RESPONSE: We removed Figure 1 and any references to it because it is not essential and in fact was not part of the original report of the results. We re-numbered Fig 2 as new Fig 1.

ED6. We note you have included a table to which you do not refer in the text of your manuscript. Please ensure that you refer to Table 1 in your text; if accepted, production will need this reference to link the reader to the Table

RESPONSE: There is only one table in the paper, Table 1, and it is referred as such in text in the original submission. We are unsure where your concern originated from, so please be more specific about what prompted it and we shall find a remedy.

COMMENT Reviewer #1: The possible relationship between atmospheric PAHs and ones in human blood. This is not clear yet, therefore this study did not conclude any relationship between bioindicator and human blood. This result is common and expected. If the authors show evidences for this unclear circumstance, it would be orijinal for the readers.

RESPONSE: We did not study association with atmospheric PAH but rather PAH in moss as indicator of air pollution. It is not obvious to us why atmospheric PAH would not contribute to PAH adducts in blood, given that this is known to occur in highly contaminated settings.

COMMENTS Reviewer#2: (REV2)

REV_01: Although inconclusive, the manuscript entitled "Association of polycyclic aromatic hydrocarbons in moss with blood biomarker among nearby residents in Portland, Oregon," is interesting and may be useful to other researchers. Yet some gaps need to be addressed before publication. 

RESPONSE: Thank you for our overall positive assessment. We attempted to address your concerns in the revision as described below in responses to your specific comments.

REV_02: One issue which needs to be addressed stems from surveys of volunteers asking about possible exposure within 48 h hours. This short period is not explained or discussed, although PAHs-adduct half-lives are estimated to be 3-4 months. The authors state that half-lives of PAHs in moss are unknown. However, one would expect, based on PAH stability, they will be at least on a month scale that is much longer than 48h hours.

RESPONSE: This is an excellent point and here we struggled with what can be reliably assessed versus what we wanted to know, reaching a compromise typical of human subjects research such as ours; we added the following explanation: “We recognize that 48 hours is shorter than half-life of PAH but it is a time period that is typically used in epidemiology to more reliably elicit person’s typical behaviors, with a recall on the order of months being unreliable; however, it is customarily assumed that behaviors in the last 48 hours are segregate persons who typical exhibit these behaviors over long periods of time versus those who do not.”

REV_03: It appears the authors omitted the description of sample preparation and analysis of DNA adducts and which specific adducts were analyzed. Alternatively, the reference to the lab performing the work should be provided if this was done as a service. 

RESPONSE: We provide only highlights of the method, with reference to exhaustive details already published and included the name of the only lab that performs this adduct assay, as stated “A licensed phlebotomist collected 50 ml of blood by venipuncture. Blood samples were transported by overnighted delivery for quantification of PAH-DNA adduct at the laboratory at Columbia University’s Mailman School of Public Health following protocol of Santella et al. [11-13].”

REV_04: Furthermore, some of the material reported in the results belong to the experimental section. The results section is surprisingly short, extended by the lengthy, simply formatted table. Nevertheless, the authors further narrowed it by focusing on only three representatives PAHs.

RESPONSE: We are unsure which of the results the reviewer wants to see in the description of methods. We aimed to keep the results short and to the point, in keeping with the small nature of the study and a rather straight-forward analysis. We avoided duplication of results in text that are best presented in a tabular format.

REV_05: The major revision of the manuscript needed to address these issues and detailed comments below.

Detailed comments

Line 19. Description of PAHs – which is a full class of compounds as "component" is awkward; please rephrase perhaps as follows "Polycyclic aromatic hydrocarbons (PAHs) are air pollutants.".

RESPONSE: Revised as suggested. Thank you.

REV_06: The direct speech in the abstract using "we" pronoun is refreshing but somewhat overused. To make articles easier to read, some sentences should be rewritten. This is also the case for other sections of the paper.

RESPONSE: We prefer to maintain active voice as it is indeed far easier on the ear than passive voice and it stresses that we, the investigators, did the work, not some abstract entity on our behalf. However, if some specific sentences appear awkward, please identify them and we will gladly attempt to improve the easy of reading.

REV_07: Line 30. It should be "adducts" in the plural.

RESPONSE: Correction made, thank you.

REV_08: Line 76. In this section, the authors carefully describe IRB protocol and data collection. The questionnaire included information of possible exposure to PAHs for 48h. Yet the question is if this is sufficient time for correlations made. The PAHs are quite stable lipophilic compounds that can bioaccumulate. Thus investigation to relationships with longer times seems to be more appropriate. Perhaps authors need to add some explanation for this or can include in the discussion how their approach is appropriate.

RESPONSE: This is a very important issue and we are glad of a chance to explain as detailed above: the 48 hours is a period over which a reliable recall of behaviors can be obtained, deemed in epidemiology to segregate, on average, persons who do or do not exhibit the behavior over longer term.

REV_09: Lines 84, 90. The symbols for degree Celsia are inconsistent; please ensure correct symbols are used everywhere. Usually, "space" is in front of the degree symbol.

RESPONSE: Correction made, thank you.

REV_10: Line 93. Use N2 for nitrogen gas.

RESPONSE: Correction made, thank you.

REV_11: Line 96. "Mass selective detector" does not need to be capitalized. Also, provide mass range and ionization – presumably electron ionization.

RESPONSE: Correction made, thank you. We believe the references to EPA methods and instruments contain all the necessary details to reproduce our work. 

REV_12: Line 101. Consider switching the order of 1st two sentences – The current presentation of the 1st sentence without specification of type of statistical analysis is confusing.

RESPONSE: We prefer to retain current order of presentation, because imputation of non-detects does not depend on statistical methods that follow.

REV_13: The experimental section does not address analysis of the PAH-adduct markers. 

RESPONSE: We are unsure what the reviewer means because association with the adducts is central to the results section. Please indicate specifically which pieces of information you seek, and we shall try to supply them.

REV_14: Some material from Lines 118-126 including figure 1 belongs to the experimental section.

RESPONSE: Persons you enrolled in the study are the result of application of our method of recruitment. Therefore, these matters are typically presented in the results in our area of scholarship; we appreciate that different conventions exist elsewhere but prefer to structure the narrative in a manner that is most familiar to us. Figure 1 was removed.

REV_15: Lines 128-131 also belong to the experimental section, supporting the study's quality but not being actual results.

RESPONSE: We moved the sentence to methods as suggested.

REV_16: Line 137-139 describes how the study was narrowed, yet correlation for other PAHs could be performed and included in the expanded figure or supplementary materials. Such data would add strength to the paper.

RESPONSE: We added results of principal components analysis as supplementary material, S1 Tables, as suggested.

REV_17: Line 140. The authors mention that specific values for concentration are not informative due to the lack of data on PAHs in moss. It seems it would be useful to compare these results to papers referred in the introduction, justifying the use of this approach.

RESPONSE: We are unsure that comparison to other papers will help in this regard because methods of analysis are far from standardized and there are no criteria values to benchmark any levels in terms of risk.

REV_18: Table 1 need significant revision to achieve a professional look, but also concerning content. PAHs naming nomenclature needs to be revised (there are numerous excellent resources on correct naming). The alphabetic order of PAHs is confusing; it would be better to organize PAH based on volatility or the number of rings and also provide their totals. Correlations with totals of these groupings could be insightful and give a different look at attempted correlation.

RESPONSE: Thank you for prompting us to review the order of PAH in the table. The correction that we made is to move 1-Methylnaphthalene and 2-Methylnaphthalene to appear right above Naphthalene, to restore alphabetical listing. We find alphabetical listing helpful and are unsure what concerns this creates; we are happy to follow specific suggestions, but we followed the spelling and order of PAH from Page 13A-2 of “Compendium of Methods for the Determination of Toxic Organic Compounds in Ambient Air Second Edition Compendium Method TO-13A Determination of Polycyclic Aromatic Hydrocarbons (PAHs) in Ambient Air Using Gas Chromatography/Mass Spectrometry (GC/MS)” published in 1999 by the US EPA (https://www.epa.gov/sites/default/files/2019-11/documents/to-13arr.pdf). Please note that EU regulations also list PAH alphabetically, but we adopted EPA naming convention/spelling (https://www.hbm4eu.eu/hbm4eu-substances/pahs/). Correlation with totals would not be appropriate in this analysis because of problems caused by non-detects (how to add them?) and the fact that there are two principal components (pointing to two independent groupings of PAHs), which makes any calculation of a total (even if possible) to be misleading.

---

## [Decision Letter · Decision Letter 1]

18 Nov 2022

PONE-D-22-13009R1Association of polycyclic aromatic hydrocarbons in moss with blood biomarker among nearby residents in Portland, OregonPLOS ONE

Dear Dr. Burstyn,

Thank you for submitting your manuscript to PLOS ONE. After careful consideration, we feel that it has merit but does not fully meet PLOS ONE’s publication criteria as it currently stands. Therefore, we invite you to submit a revised version of the manuscript that addresses the points raised during the review process.

Your manuscript revision has now been reviewed, and the reviewers still proposed a few amendments.  Having looked at your paper myself, I agree with the reviewers that it needs some minor work on the nomenclature of PAHs and restructuring by increasing the discussion regarding the no correlation of PAH levels in moos and PAH-DNA adducts in human blood, which seems to make more sense, and would likely attract a wider readership.

You are requested to consider these carefully and to prepare a revised version of the paper.  After your amendments, please return the manuscript to me via the system, and we will reconsider it for publication.  Please ensure you provide a revised version of the manuscript along with any Figures and Tables.

We look forward to receiving your revised manuscript.

Kind regards,

Fung-Chi Ko, PhD

Academic Editor

PLOS ONE

Journal Requirements:

Additional Editor Comments:

Reviewers' comments:

Reviewer #1:  PAHs are important air pollutants with very common sources. Many studies are being conducted to establish a relationship with the personal exposures of these pollutants. While it is difficult to establish a statistical relationship between PAHs and human disease even in active sampling and personal exposure studies, it is much more difficult to demonstrate this relationship with bioindicators. (if possible, more discussion is strongly recommended to reveal the relationship).

Reviewer #2:  The authors addressed the majority of comments, I have just a couple of comments:

I still disagree with naming nomenclature. The EPA is agency providing regulations, limits, and guidelines, not nomenclature. The letters in brackets should be in italics, and the brackets should be square. For details, see NIST reference https://nvlpubs.nist.gov/nistpubs/Legacy/SP/nistspecialpublication922.pdf

Line 101.I presume it should be “Restek” column.”

Reviewer #3:  Algae are used to determine PAH levels in outdoor air, but not enough to compare this with human DNA. As you mentioned, it is costly to examine the external environment with classical methods. For this reason, it is being investigated which plant species to use or not to use to determine the atmospheric levels of persistent organic pollutants. The results section is very short.

Reviewer's Responses to Questions

**Comments to the Author**

1. If the authors have adequately addressed your comments raised in a previous round of review and you feel that this manuscript is now acceptable for publication, you may indicate that here to bypass the “Comments to the Author” section, enter your conflict of interest statement in the “Confidential to Editor” section, and submit your "Accept" recommendation.

Reviewer #2: (No Response)

Reviewer #3: (No Response)

2. Is the manuscript technically sound, and do the data support the conclusions?

Reviewer #2: Yes

Reviewer #3: (No Response)

3. Has the statistical analysis been performed appropriately and rigorously? 

Reviewer #2: Yes

Reviewer #3: (No Response)

4. Have the authors made all data underlying the findings in their manuscript fully available?

Reviewer #2: Yes

Reviewer #3: (No Response)

5. Is the manuscript presented in an intelligible fashion and written in standard English?

Reviewer #2: Yes

Reviewer #3: (No Response)

6. Review Comments to the Author

Reviewer #2: The authors addressed the majority of comments, I have just a couple of comments:

I still disagree with naming nomenclature. The EPA is agency providing regulations, limits, and guidelines, not nomenclature. The letters in brackets should be in italics, and the brackets should be square. For details, see NIST reference https://nvlpubs.nist.gov/nistpubs/Legacy/SP/nistspecialpublication922.pdf

Line 101.I presume it should be “Restek” column.”

Reviewer #3: Algae are used to determine PAH levels in outdoor air, but not enough to compare this with human DNA. As you mentioned, it is costly to examine the external environment with classical methods. For this reason, it is being investigated which plant species to use or not to use to determine the atmospheric levels of persistent organic pollutants. The results section is very short. For this reason, I could not find a result that could change my initial thought. The authors described their work poorly. For this reason, I do not consider it a work worth publishing.

7. PLOS authors have the option to publish the peer review history of their article (what does this mean?). If published, this will include your full peer review and any attached files.

Reviewer #2: No

Reviewer #3: No

---

## [Author Response · Author response to Decision Letter 1]

29 Nov 2022

RESPONSE: 

We appreciate a chance to consider further revisions and respond to reviewers. We are glad that that there are no specific concerns articulated by the editor and that none of the reviewers appear to identify problems with our methodology. Specific responses to comments and suggestions are listed below, following the reviewers’ comments. We now hope that the manuscript is acceptable for publication but are open to further recommendations for improvement.

Reviewer #1: PAHs are important air pollutants with very common sources. Many studies are being conducted to establish a relationship with the personal exposures of these pollutants. While it is difficult to establish a statistical relationship between PAHs and human disease even in active sampling and personal exposure studies, it is much more difficult to demonstrate this relationship with bioindicators. (if possible, more discussion is strongly recommended to reveal the relationship).

RESPONSE: We added sentence conveying suggested meaning to the conclusion: “Although we do not challenge the established observation that PAH in environment can cause adverse health effects and produce adducts, our results indicate that it is difficult to demonstrate this relationship with bioindicators in the studied setting. It is plausible that in other environments with a stronger relationship between personal exposure and contamination of properties by PAH bioindicators will prove advantageous over established methods of assessing risk and exposure to humans.”.

Reviewer #2: The authors addressed the majority of comments, I have just a couple of comments:

I still disagree with naming nomenclature. The EPA is agency providing regulations, limits, and guidelines, not nomenclature. The letters in brackets should be in italics, and the brackets should be square. For details, see NIST reference https://nvlpubs.nist.gov/nistpubs/Legacy/SP/nistspecialpublication922.pdf

RESPONSE: We changed brackets to square and the letters in brackets to italics. We trust that the journal will have the final say on the house style.

Line 101.I presume it should be “Restek” column.”

RESPONSE: Correction made; apologies for the typo.

Reviewer #3: Algae are used to determine PAH levels in outdoor air, but not enough to compare this with human DNA. As you mentioned, it is costly to examine the external environment with classical methods. For this reason, it is being investigated which plant species to use or not to use to determine the atmospheric levels of persistent organic pollutants. The results section is very short.

RESPONSE: We agree with the reviewers views on cost of traditional methods of accessing personal exposure to PAH in general environment and were disappointed that bioindicators were not useful as an alternative in the studied setting. We also agreed that this does not mean that bioindicators cannot be useful in other settings and added two sentences to the conclusions to emphasize this: “Although we do not challenge the established observation that PAH in environment can cause adverse health effects and produce adducts, our results indicate that it is difficult to demonstrate this relationship with bioindicators in the studied setting. It is plausible that in other environments with a stronger relationship between personal exposure and contamination of properties by PAH bioindicators will prove advantageous over established methods of assessing risk and exposure to humans.”. We pride ourselves on construction laconic arguments, hence a short exposition of the results, which speak for themselves through figure and table. Since the reviewer did not specify what they want us to further comment on in the results, we are unable to act on the observation that the section is “very short”.

---

## [Editor Report · Decision Letter 2]

2 Dec 2022

Association of polycyclic aromatic hydrocarbons in moss with blood biomarker among nearby residents in Portland, Oregon

PONE-D-22-13009R2

Dear Dr. Burstyn,

We’re pleased to inform you that your manuscript has been judged scientifically suitable for publication and will be formally accepted for publication once it meets all outstanding technical requirements.

Kind regards,

Fung-Chi Ko, PhD

Academic Editor

PLOS ONE

---

## [Editor Report · Acceptance letter]

6 Dec 2022

PONE-D-22-13009R2 

Association of polycyclic aromatic hydrocarbons in moss with blood biomarker among nearby residents in Portland, Oregon 

Dear Dr. Burstyn:

I'm pleased to inform you that your manuscript has been deemed suitable for publication in PLOS ONE. Congratulations! Your manuscript is now with our production department. 

Kind regards, 

on behalf of

Dr. Fung-Chi Ko 

Academic Editor

PLOS ONE